# AKAP150-anchored PKA regulates synaptic transmission and plasticity, neuronal excitability and CRF neuromodulation in the mouse lateral habenula
Sarah C. Simmons[1,3], William J. Flerlage[1,3], Ludovic D. Langlois[1], Ryan D. Shepard[1], Christopher Bouslog[1], Emily H. Thomas[1], Kaitlyn M. Gouty[1], Jennifer L. Sanderson[2], Shawn Gouty[1], Brian M. Cox [1], Mark L. Dell'Acqua [2] ✉ & Fereshteh S. Nugent [1] ✉

The scaffolding A-kinase anchoring protein 150 (AKAP150) is critically involved in kinase and phosphatase regulation of synaptic transmission/plasticity, and neuronal excitability. Emerging evidence also suggests that AKAP150 signaling may play a key role in brain's processing of rewarding/aversive experiences, however its role in the lateral habenula (LHb, as an important brain reward circuitry) is completely unknown. Using whole cell patch clamp recordings in LHb of male wildtype and ΔPKA knockin mice (with deficiency in AKAP-anchoring of PKA), here we show that the genetic disruption of PKA anchoring to AKAP150 significantly reduces AMPA receptor-mediated glutamatergic transmission and prevents the induction of presynaptic endocannabinoid-mediated long-term depression in LHb neurons. Moreover, ΔPKA mutation potentiates GABA_A receptor-mediated inhibitory transmission while increasing LHb intrinsic excitability through suppression of medium afterhyperpolarizations. ΔPKA mutation-induced suppression of medium afterhyperpolarizations also blunts the synaptic and neuroexcitatory actions of the stress neuromodulator, corticotropin releasing factor (CRF), in mouse LHb. Altogether, our data suggest that AKAP150 complex signaling plays a critical role in regulation of AMPA and GABA_A receptor synaptic strength, glutamatergic plasticity and CRF neuromodulation possibly through AMPA receptor and potassium channel trafficking and endocannabinoid signaling within the LHb.

The scaffold protein A-kinase anchoring protein 79/150 (AKAP150, 79 human/150 rodent/*Akap5* gene) is a crucial regulator of synaptic receptor trafficking, synaptic transmission and plasticity, and neuronal excitability by anchoring protein kinases (e.g., protein kinase A, PKA, and protein kinase C, PKC) and phosphatases (e.g., calcineurin, CaN) and other signaling molecules (e.g., the transcription factor nuclear factor of activated T-cells, NFAT) to subcellular nanodomains at specific synapses (glutamatergic and GABAergic)[1–9] and ion channels [e.g., M-type potassium channels, A-type potassium channels, transient receptor potential vanilloid 1 and L-type calcium channels][10–16]. AKAP150 anchoring of PKA, PKC and CaN has been shown to mediate the opposing effects of these enzymes in post-synaptic trafficking of both AMPA receptors (AMPARs) and GABA_A receptors (GABA_ARs) during glutamatergic and GABAergic plasticity[1–9]. For example, phosphorylation of GluA1 subunit of AMPARs with

[1]Uniformed Services University of the Health Sciences, Department of Pharmacology and Molecular Therapeutics, Bethesda, MD 20814, USA. [2]Department of Pharmacology, University of Colorado School of Medicine, Anschutz Medical Campus, Aurora, CO 80045, USA. [3]These authors contributed equally: Sarah C. Simmons, William J. Flerlage. ✉e-mail: mark.dellacqua@cuanschutz.edu; fereshteh.nugent@usuhs.edu

AKAP150-anchored PKA is required for stabilization and insertion of AMPARs in the synapse and promotion of long-term potentiation, while AKAP-anchored CaN dephosphorylates GluA1 subunit of AMPARs and removes AMPARs from the synapse which is required for long-term depression (LTD) in hippocampal CA1 neurons[17]. Similarly, transient recruitment of GluA2-lacking calcium permeable AMPARs (CP-AMPARs) through phosphorylation coordinated by AKAP150/PKA/CaN is required along with NMDA receptors (NMDARs) not only for the induction of input-specific long-term potentiation and LTD but also for homeostatic plasticity (synaptic scaling) in the hippocampus[9,18,19].

In spite of the increasing in-depth mechanistic insights into the role of AKAP150 complex in hippocampal-related learning and memory processes, less is known about the normal and pathological roles of AKAP150 complex-dependent signaling in neural processes within reward-related brain circuits that could contribute to reward-related behaviors as well as in the development of neurological and neuropsychiatric illnesses. This is an important area of psychiatric research as human studies of polymorphisms of *AKAP5* also indicate that individuals carrying *AKAP5* polymorphisms show altered emotional processing and behavioral responses including aggression, expression of anger and impulsivity associated with alterations in the function in limbic regions[20–22]. Moreover, copy number variations in *AKAP5* have been found in DNA samples of schizophrenia patients but not in control subjects[23], suggesting the possible involvement of *AKAP5* in the pathogenesis of schizophrenia, a neurodevelopmental disorder also linked to reward circuit dysfunction and high rates of addiction[24–26]. Consistently, in recent years a few studies highlighted the importance of AKAP150 signaling within the ventral tegmental area (VTA)[8,27–29], nucleus accumbens[30–32], amygdala[33,34] and periaqueductal gray[15] in regulation of synaptic plasticity and the development of depressive states, aversive and drug-related behaviors.

Here, we attempted to address the potential impact of AKAP150-anchored PKA on the lateral habenula (LHb); an anti-reward brain region hub that regulates midbrain monoaminergic centers and is involved in reward/motivation, mood regulation and decision-making. Accumulating evidence indicate that LHb hyperactivity plays an instrumental role in pathophysiology of depression and possibly other mood disorders and substance use disorders, thus LHb is gaining interest as a potential target for neuromodulation and antidepressants[35–38]. LHb neurons are excited by aversive and unpleasant events or the absence of expected reward, and inhibited by unexpected reward, encoding behavioral avoidance and reward prediction errors through suppression of VTA dopamine and dorsal raphe nucleus serotonin systems[35,36]. The majority of LHb neurons are believed to be glutamatergic and long-range projecting, although local glutamatergic

and GABAergic connections within the LHb are reported[39–42]. LHb neurons receive glutamatergic, GABAergic and co-releasing glutamate/GABA inputs from the basal ganglia and diverse limbic areas including medial prefrontal cortex, entopeduncular nucleus, lateral preoptic area, lateral hypothalamus, ventral pallidum, medial and lateral septum, central amygdala, bed nucleus of stria terminals as well as receiving reciprocal inputs from the VTA and periaqueductal gray. LHb projects to the substantia nigra, VTA, rostromedial tegmental area (RMTg), dorsal raphe nucleus, locus coeruleus and periaqueductal gray[35,43]. The majority of the glutamatergic output of LHb exerts a potent feedforward inhibitory influence on monoaminergic systems including VTA dopamine neuronal activity by excitation of GABAergic interneurons and of GABAergic neurons of the RMTg[44–47].

LHb hyperactivity is found to be a common finding associated with anhedonia, lack of motivation and social withdrawal which reflect some of the core features of reward deficits seen in clinical depression[36,48–50]. In general, LHb dysfunction can mediate negative affective states, social deficits, risky decision-making and impulsivity (as shown in patients with depression, schizophrenia, Parkinson's disease and attention-deficit hyperactivity disorder)[36,48–56]. Given the known postsynaptic PKA-mediated control of LHb synaptic function and intrinsic excitability by neuromodulatory actions of corticotropin releasing factor (CRF)-CRF receptor 1 signaling that acts via cAMP and PKA[57], here we investigated the potential impact of AKAP150-anchored PKA on LHb synaptic and neuronal function and CRF neuromodulation within the LHb using whole-cell patch clamp recordings and AKAP150ΔPKA knockin mouse model (hereafter referred to as ΔPKA mice)[14]. The ΔPKA mice have an internal deletion of ten amino acids within the PKA-RII subunit binding domain near the AKAP C terminus. For AKAP150 complex-related studies, they are advantageous compared to AKAP150 knockout mice as the mutation only affects AKAP150-anchoring of PKA without disrupting other AKAP150 interactions[14]. We found that the genetic disruption of PKA anchoring to AKAP150 significantly altered both AMPAR- and GABA$_A$R-mediated synaptic transmission and impaired the induction of an endocannabinoid (eCB)-mediated LTD in LHb neurons. Moreover, we observed that ΔPKA mutation enhanced LHb intrinsic excitability which then blunted the excitatory effects of CRF on LHb neuronal activity. Given the multifaceted impact of AKAP150 anchoring of PKA in regulation of glutamatergic transmission and plasticity and neuronal excitability of LHb as well as alteration of CRF regulation of LHb excitability, our data suggest a key role for the AKAP150 complex in normal LHb function and potential contributions of defective AKAP150-mediated PKA anchoring to aberrant LHb activity, dysregulation of CRF neuromodulation within LHb circuits, and hence mood dysregulation.

## Results

### AKAP150 expression in the LHb
Figure 1 depicts a representative 40x image of LHb of a young adult male mouse taken at AP location (−1.34 relative to bregma). We observed wide expression of AKAP150 in the LHb at three AP locations (−1.06, −1.34 and −1.46) in 4 wild type (WT) mice.

### Effects of AKAPΔPKA mutation on synaptic transmission and glutamatergic LTD in LHb neurons
To examine the effects of genetic disruption of PKA anchoring to AKAP150 on AMPAR and GABA$_A$R-mediated synaptic transmission, we recorded either mEPSCs (Fig. 2) or mIPSCs (Fig. 3) from LHb neurons from WT and ΔPKA mice with deficiency in PKA-anchoring to AKAP150[14]. ΔPKA mutation significantly decreased the average amplitude (inset in Fig. 2b), frequency (inset in Fig. 2c) and charge transfer (inset in Fig. 2d) of mEPSCs and correspondingly shifted the cumulative probability curves of mEPSC amplitude (to the left indicative of smaller amplitude, Fig. 2b), inter-event interval (IEI, to the right indicative of lower frequency, Fig. 2c) and charge transfer (to the left indicative of lower charge transfer, Fig. 2d) without altering mEPSC tau decay (Fig. 2, two-tailed unpaired Student's *t*-tests, Kolmogorov-Smirnov tests, $p < 0.05$, $p < 0.01$, $p < 0.001$, $p < 0.0001$),

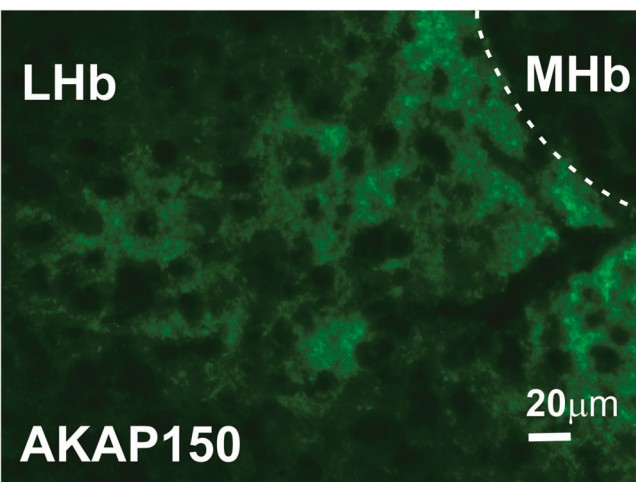

**Fig. 1 | AKAP150 is expressed in the LHb.** Example of a brain section stained with antibody to AKAP150 (green), showing the expression of AKAP150 in the mouse LHb. Scale bar, 20 μm.

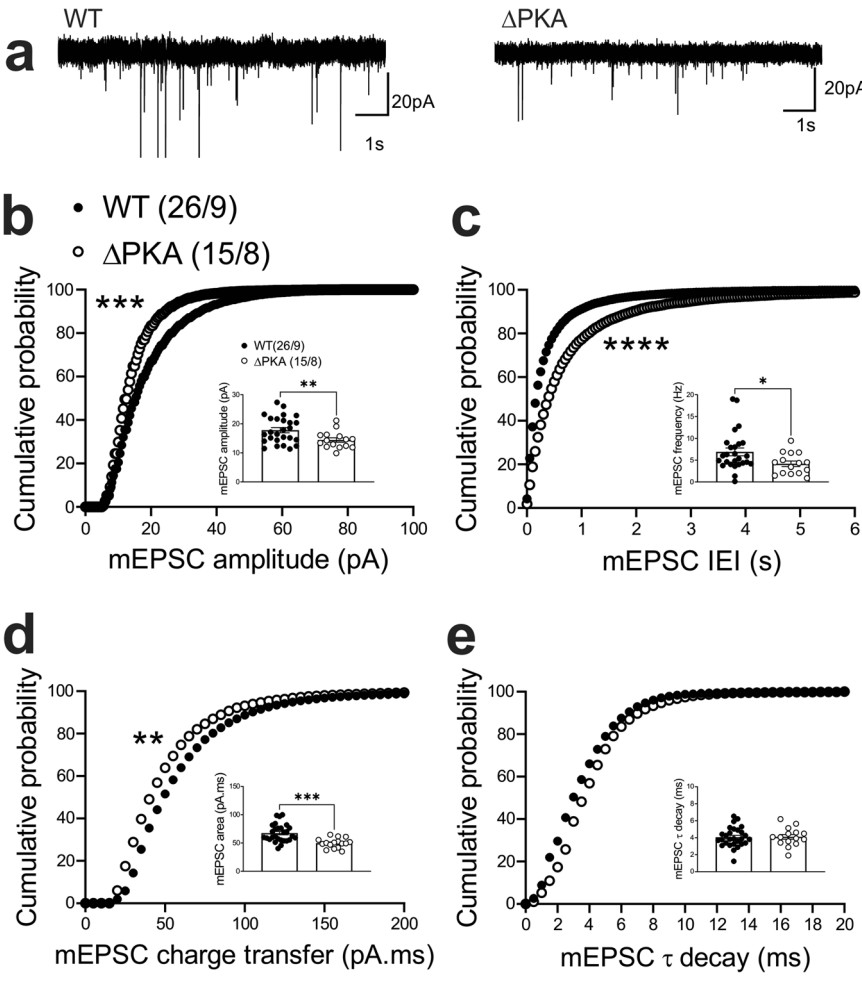

**Fig. 2 | Genetic disruption of AKAP150-anchored PKA depressed glutamatergic transmission in LHb neurons. a** Sample AMPAR-mediated mEPSC traces from WT (left) and ΔPKA mice (calibration bars: 20 pA/1 s). Average bar graphs of mEPSC and cumulative probability plots of (**b**) amplitude, (**c**) frequency (inter-event interval), (**d**) charge transfer and (**e**) τ decay for all mEPSCs in WT (filled symbols, $n = 26$ cells from 9 mice) and ΔPKA (open symbols, $n = 15$ cells from 8 mice). Two-tailed unpaired Student's $t$-tests and Kolmogorov-Smirnov tests, *$p < 0.05$, **$p < 0.01$, ***$p < 0.001$, ****$p < 0.0001$. In this and all the following figures, the average data are presented as mean ± SEM.

suggesting both pre- and postsynaptic suppression of glutamatergic transmission in LHb neurons. Only the cumulative probability curve of mIPSC amplitude (Fig. 3b) was significantly shifted to the right by this genetic AKAP-PKA disruption, which may indicate an increase in postsynaptic GABA$_A$R function at a subset of GABAergic synapses onto LHb neurons (Fig. 3, Kolmogorov-Smirnov tests, $p < 0.0001$). All other mIPSC properties were not significantly different between ΔPKA and WT.

Previously, it has been reported that low frequency stimulation (LFS) can induce a retrograde presynaptic eCB-mediated LTD of the AMPAR-mediated electrically-evoked EPSCs in LHb neurons by postsynaptic activation of group I metabotropic glutamate receptors[58] or through calcium-permeable AMPARs (CP-AMPARs) that further activate NMDARs[59,60]. Here, we also used an identical LFS protocol to induce a presynaptic LTD and assessed the paired pulse ratios (PPRs) and the inverse square of the coefficient of variation (CV = SD/mean; 1/CV$^2$) values as the two main indicators of presynaptic expression of synaptic plasticity. As shown in Fig. 4, the LTD protocol strongly induced eCB-LTD onto LHb neurons in WT mice (Figs. 4a, c) which was associated with significant increases in PPRs (Fig. 4d) and corresponding decreases in 1/CV$^2$ (Fig. 4e) suggesting the expression of eCB-mediated LTD. On the other hand, LHb neurons from ΔPKA mice (Figs. 4b, c) were unable to express this retrograde presynaptic glutamatergic LTD (Fig. 4, LTD: WT, $F (1.53, 6.67) = 16.76$; ΔPKA: $F (1.32, 4.24) = 1.04$, $p = 0.39$, Mixed-effects ANOVA. PPRs and I/CV$^2$: unpaired Student's $t$-test, $p < 0.01$, $p < 0.05$).

### Effects of AKAPΔPKA mutation on LHb intrinsic excitability
Given that PKA regulation of various ion channels in the neuronal membrane requires anchoring of PKA by AKAP150[61], it was possible that the

genetic disruption of PKA-AKAP association in ΔPKA mice could also impact intrinsic plasticity through changes in trafficking and/or function of a number of voltage-gated channels. Consistently, we observed that LHb neurons of ΔPKA mice exhibited significantly higher intrinsic excitability in the absence of synaptic transmission compared to those from WT mice (Fig. 5a). Furthermore, ΔPKA mutation-induced intrinsic plasticity was associated with lower amplitude of medium afterhyperpolarizations (mAHPs) (Fig. 5d), and shorter AP half widths (Fig. 5g) suggesting that the genetic disruption of AKAP150-PKA anchoring modified intrinsic active and passive neuronal membrane properties, which could also influence synaptic conductance (Fig. 5a–g, intrinsic excitability: $F (1, 209) = 21.22$, 2-way ANOVA; mAHPs and AP half-width: unpaired Student's $t$ test, $p < 0.05$, $p < 0.01$, $p < 0.0001$).

### Effects of AKAPΔPKA mutation on CRF neuromodulation within the LHb
Previously, we demonstrated that the LHb is a highly CRF-responsive brain region with PKA-dependent regulation of LHb synaptic inhibition and intrinsic excitability. We showed that CRF acting through postsynaptic CRF receptor 1 and cAMP-PKA signaling increases LHb excitability through PKA-dependent suppression of small conductance potassium SK channel activity, as well as presynaptic GABA release via retrograde eCB-CB1 receptor signaling in rat LHb neurons without any significant alterations in glutamatergic transmission[57]. In contrast to our earlier findings in rat LHb where exogenous CRF did not alter mEPSCs, CRF bath application significantly decreased the average frequency of mEPSCs (inset in Fig. 6c) and correspondingly shifted the cumulative probability curves of mEPSC IEI to the right (Fig. 6c) (Fig. 6, paired Student's $t$-tests, Kolmogorov-Smirnov

**Fig. 3 | Genetic disruption of AKAP150-anchored PKA potentiated GABAergic transmission in LHb neurons. a** Sample GABA$_A$R-mediated mIPSC traces from WT (left) and ΔPKA mice (calibration bars: 20 pA/1 s). Average bar graphs of mIPSC and cumulative probability plots of (**b**) amplitude, (**c**) frequency (inter-event interval), (**d**) charge transfer and (**e**) τ decay for all mIPSCs in WT (filled symbols, $n = 20$ cells from 6 mice) and ΔPKA (open symbols $n = 22$ cells from 7 mice). Two-tailed Kolmogorov-Smirnov tests, ****$p < 0.0001$.

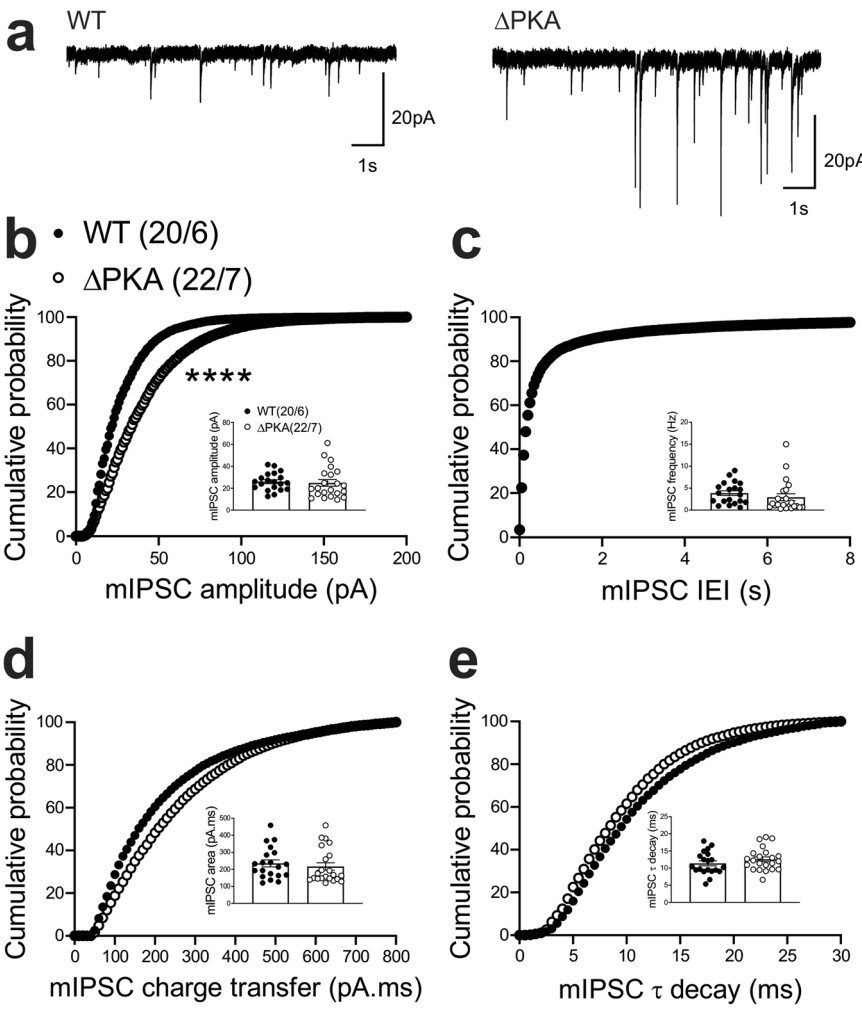

tests, $p < 0.05$, $p < 0.0001$), suggesting of CRF-induced suppression of presynaptic glutamate release in mouse LHb neurons.

This diminishing effect of CRF on presynaptic glutamate release was absent in LHb neurons of ΔPKA mice and we even detected a slight but significant leftward shift in the cumulative probability curves of mEPSC IEI (Fig. 7c) that may suggest an unmasked potentiating effects of CRF on presynaptic glutamate release upon disruption of AKAP150-PKA association (Fig. 7, Kolmogorov-Smirnov tests, $p < 0.05$).

In contrast, we observed similar effects of the ΔPKA mutation on GABAergic transmission in mouse LHb and rat LHb; exogenous CRF significantly diminished the average frequency of mIPSCs (inset in Fig. 8c) and resulted in a significant shift in the cumulative probability curves of mIPSC IEI to the right (Fig. 8c) (Fig. 8, paired Student's $t$-tests, Kolmogorov-Smirnov tests, $p < 0.05$, $p < 0.0001$), indicating CRF-induced suppression of presynaptic GABA release onto mouse LHb neurons. This diminishing effect of CRF on presynaptic GABAergic transmission remained intact in LHb neurons of ΔPKA mice as evident with a smaller but still significant rightward shift in the cumulative probability curves of mIPSC IEI (Fig. 9c). However, CRF additionally decreased the average amplitude (inset in Fig. 9b) and charger transfer (inset in Fig. 9d) of mIPSCs as well as shifted their corresponding cumulative probability curves of mIPSC amplitude (Fig. 9b) and charge transfer (Fig. 9d) to the left. These results may indicate that CRF alters postsynaptic function and/or trafficking of GABA$_A$Rs upon disruption of AKAP150-PKA association, potentially favoring AKAP150-dependent regulation of anchored PKC and/or CaN phosphatase activity, both of which can negatively regulate postsynaptic GABA$_A$Rs[29,62], downstream of CRF-CRFR-PKC signaling[63,64] (Fig. 9, paired Student's $t$-tests, Kolmogorov-Smirnov tests, $p < 0.05$, $p < 0.0001$).

In a subset of the neurons represented in Fig. 5a, we were also able to examine the effects of CRF bath application on LHb intrinsic excitability in slices from WT and ΔPKA mice (i.e., intrinsic excitability recordings before and after CRF bath application). Similar to our earlier findings in rat LHb[57], CRF bath application exerted similar effects on intrinsic excitability and intrinsic membrane properties of male mouse LHb. We found that CRF significantly increased LHb intrinsic excitability (with blocked fast AMPAR, NMDAR and GABA$_A$R-mediated transmission) (Fig. 10a, b) coincident with higher input resistance (Fig. 10c), reduced levels of mAHPs (Fig. 10e), lower AP threshold (Fig. 10f) and smaller AP amplitudes (Fig. 10g) in LHb neurons. The only exception was that CRF-induced increases in fAHPs were not observed in mouse LHb (Fig. 10d) (Fig. 10a–h, 2-way repeated-measures ANOVA, $F_{(1, 7)} = 8.23$, $p < 0.05$). On the other hand, we observed that the excitatory actions of CRF in LHb of ΔPKA mice were absent (Fig. 10i–j), although CRF was able to increase the amplitude of fAHPs (Fig. 10l) in LHb neurons of ΔPKA mice (Fig. 10i–p, 2-way repeated-measures ANOVA, $F_{(1, 6)} = 0.4436$, $p = 0.53$).

## Discussion

Most of our understanding of the role of AKAP150 in the brain relates to hippocampal studies but emerging evidence also suggests important roles for AKAP150 in reward-related brain regions critical for the control of mood, motivation, reward, and stress responses[8,15,27–31,33,34]. Here, we uncovered a multifaceted essential regulatory role of AKAP150 in synaptic function, neuronal activity, and CRF neuromodulatory actions within the LHb using the ΔPKA knock-in mouse model with a deficiency of AKAP150 anchoring of PKA. We found that in LHb neurons AKAP150-anchored PKA is required for postsynaptic regulation of AMPAR trafficking and/or

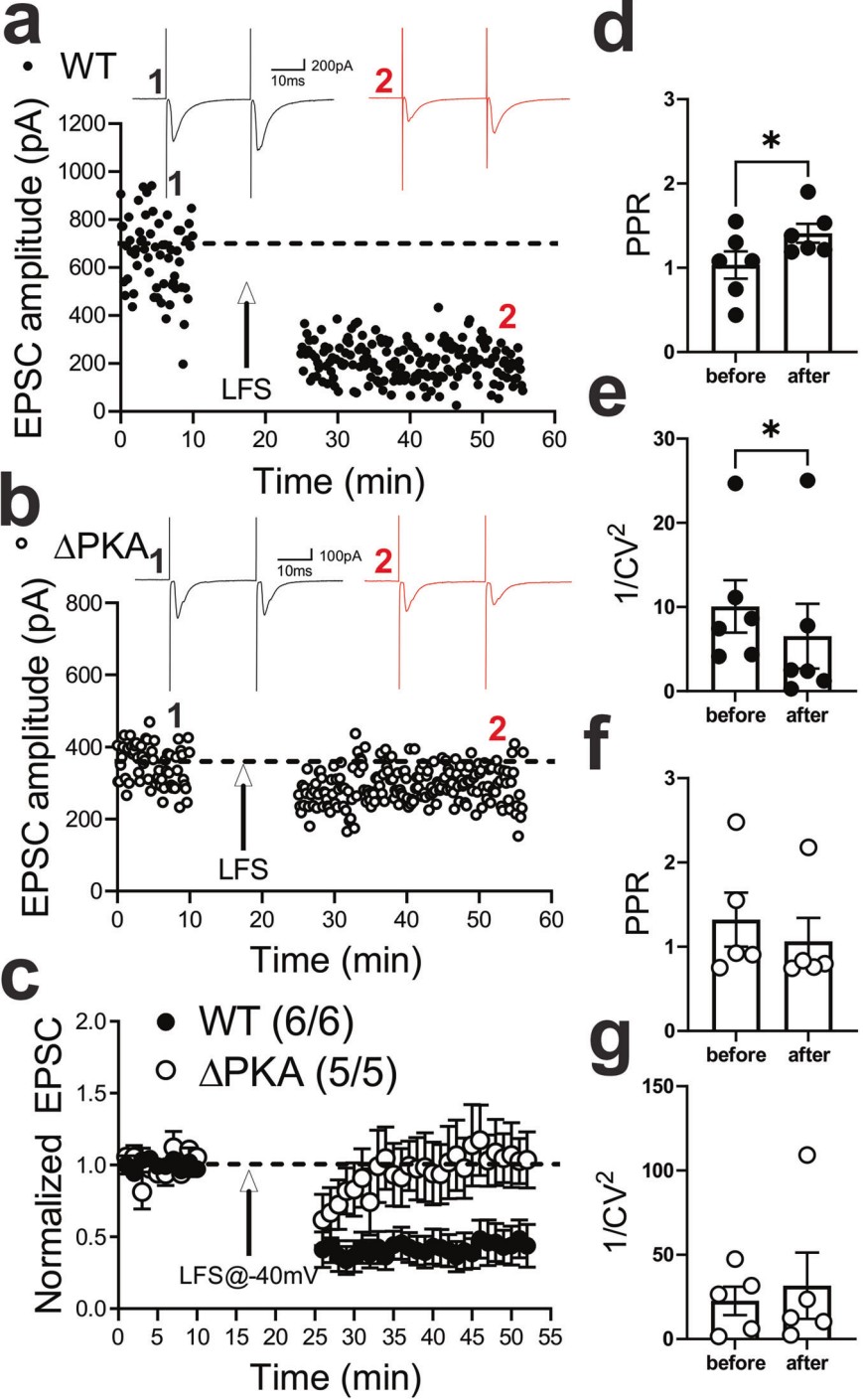

**Fig. 4 | Genetic disruption of AKAP150-anchored PKA impaired eCB-LTD induction in LHb neurons. a, b** Single experiments showing induction of LTD recorded in LHb neurons from WT (**a**) and ΔPKA (**b**) mice. At the arrow, LTD was induced using LFS while cells were depolarized at −40 mV. Insets: averaged EPSCs before (black, labeled as **1**) and 25 min after LFS (red, labeled as **2**). In this and all figures, ten consecutive traces from each condition were averaged for illustration as inset. Calibration: 100–200 pA, 10 ms. **c** Average LTD experiments with corresponding PPRs (**d, f**) and $1/CV^2$ (**e, g**) in LHb neurons recorded from WT (filled symbols, $n = 6$ cells from 6 mice) and ΔPKA (open symbols $n = 5$ cells from 5 mice). Mixed-effects ANOVA and two-tailed unpaired Student's $t$-test, $*p < 0.05$, $*p < 0.01$).

function at glutamatergic synapses and for the expression of glutamatergic retrograde presynaptic eCB-LTD. In contrast, AKAP150-PKA signaling may provide an inhibitory feedback mechanism for postsynaptic trafficking and/or function of $GABA_ARs$ or regulate gene expression programs that indirectly control inhibitory synaptic strength[65]. Moreover, we found that defects in AKAP150-PKA mediated expression, trafficking, and/or gating of potassium channels that regulate LHb excitability could blunt CRF neuromodulatory effects within the LHb.

Postsynaptic AMPAR and $GABA_AR$ trafficking and function are necessary for maintaining basal synaptic transmission as well as induction and expression of synaptic plasticity which can be altered through phosphorylation-dephosphorylation processes that require AKAP150[61]. There are four subunits of AMPARs (GluA1–GluA4) of which LHb

neurons express high levels of GluA1-containing rectifying AMPARs that lack GluA2 (also called calcium-permeable, CP-AMPARs with fast kinetics, high conductance and strong inward rectification) but also express low levels of both GluA2-containing AMPARs (with slower kinetics, low conductance and impermeability to calcium) and NMDARs at their glutamatergic synapses[66,67]. AKAP150-anchored PKA is shown to phosphorylate Ser-845 on the GluA1 subunit of CP-AMPARs to increase membrane trafficking of AMPARs at glutamatergic synapses, while AKAP150-anchored PKC, through phosphorylation of Ser-831 on GluA1 further results in emergence of CP-AMPARs at synapses[68]. In the LHb, it has been shown that cocaine can induce synaptic potentiation and hyperexcitability in LHb neurons projecting to RMTg (LHb→RMTg neurons) through Ser-845 phosphorylation of GluA1 that increases trafficking of GluA1 AMPARs in

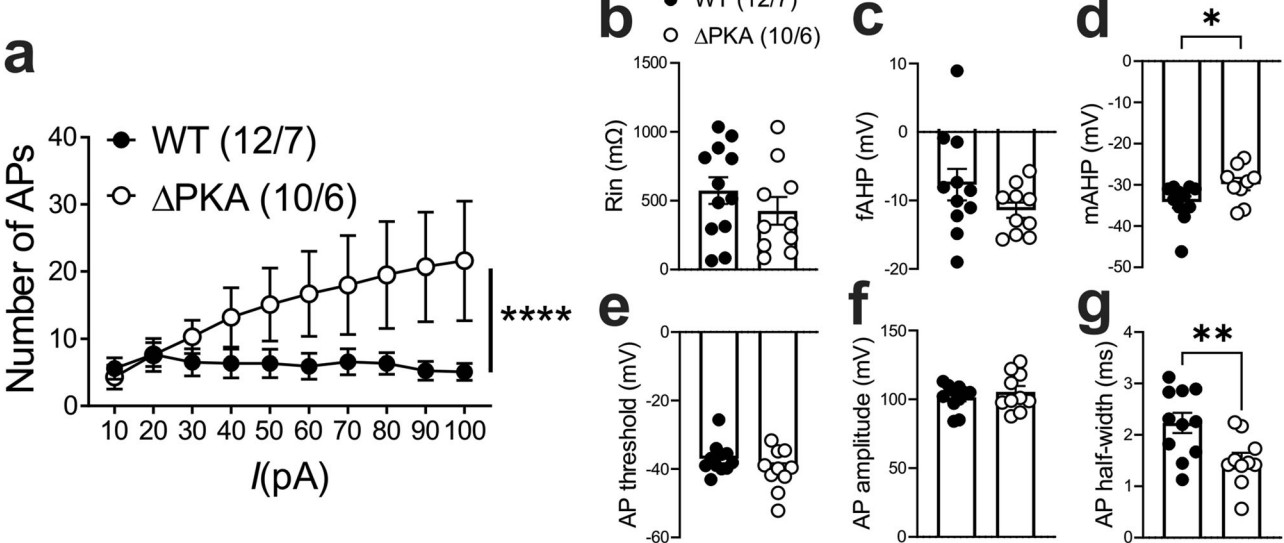

**Fig. 5 | Genetic disruption of AKAP150-anchored PKA increased LHb intrinsic excitability.** All recordings in this graph were performed with fast synaptic transmission blocked. **a–g** AP recordings in response to depolarizing current steps and corresponding measurements of Rin, fAHP, mAHP, AP threshold, AP amplitude and AP half-width in LHb neurons from WT (black filled symbols, $n = 12$ cells from 7 mice) and ΔPKA (black open symbols, $n = 10$ cells from 6 mice). 2-way ANOVA and two-tailed unpaired Student's $t$ test, *$p < 0.05$, *$p < 0.01$, ****$p < 0.0001$).

**Fig. 6 | CRF decreased presynaptic glutamate release in the LHb of WT mice. a** Sample AMPAR-mediated mEPSC traces from WT mouse before (black) and after CRF application (red, 250 nM) (calibration bars: 20 pA/1 s). Average bar graphs of mEPSC and cumulative probability plots of (**b**) amplitude, (**c**) frequency (inter-event interval), (**d**) charge transfer and (**e**) τ decay for all mEPSCs in WT mice before (black filled symbols) and after CRF (red filled symbols) ($n = 6$ cells from 4 mice). Two-tailed paired Student's $t$-tests and Kolmogorov-Smirnov tests, *$p < 0.05$, ****$p < 0.0001$.

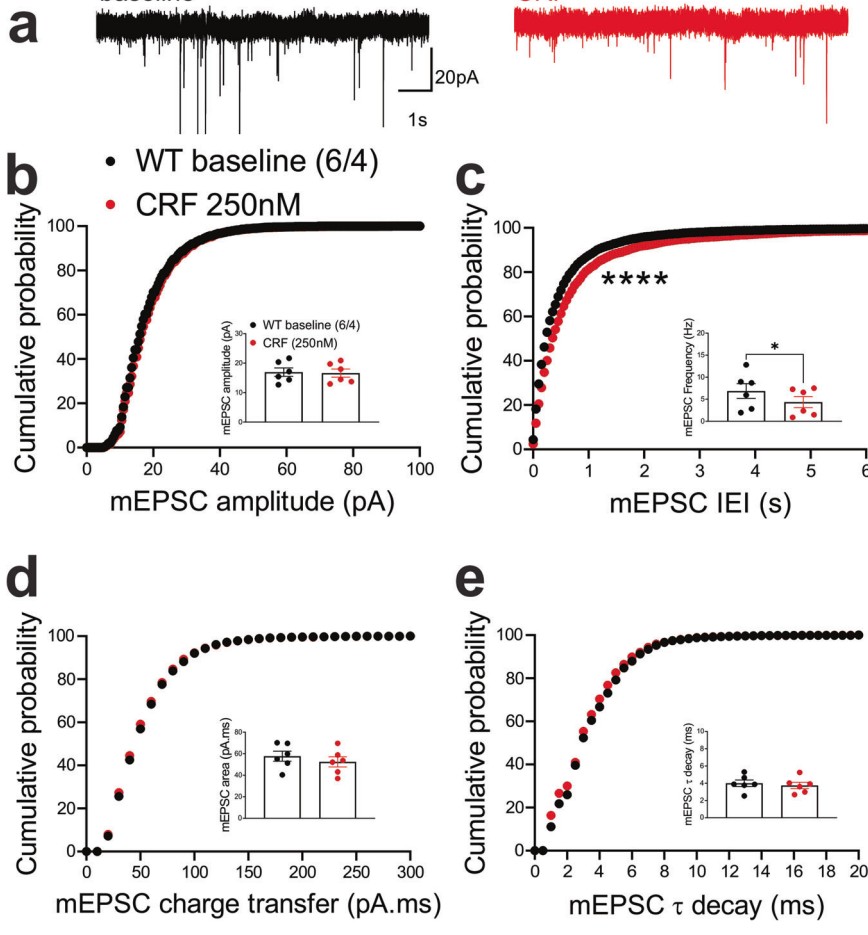

LHb$^{\rightarrow RMTg}$ neurons[69]. Additionally, phosphorylation of Ser-831 on the GluA1 subunit of AMPARs by β-calcium/calmodulin-dependent kinase type II in the LHb is shown to promote GluA1 AMPAR insertion into synapses and glutamatergic potentiation, resulting in LHb hyperactivity and behavioral depression[70]. Given that the knockin mutation in ΔPKA mice leads to reductions in postsynaptic PKA localization in dendritic spines[14], our observation of lower levels of mEPSC amplitude and charge transfer in ΔPKA mice is most likely due to decreased PKA-dependent Ser-845

**Fig. 7 | CRF slightly potentiated presynaptic glutamate release in the LHb of ΔPKA mice. a** Sample AMPAR-mediated mEPSC traces from ΔPKA mouse before (black) and after CRF application (red, 250 nM) (calibration bars: 20 pA/1 s). Average bar graphs of mEPSC and cumulative probability plots of (**b**) amplitude, (**c**) frequency (inter-event interval), (**d**) charge transfer and (**e**) τ decay for all mEPSCs in ΔPKA mice before (black open symbols) and after CRF (red open symbols) (*n* = 5 cells from 5 mice). Two-tailed Kolmogorov-Smirnov tests, *\*p* < 0.05.

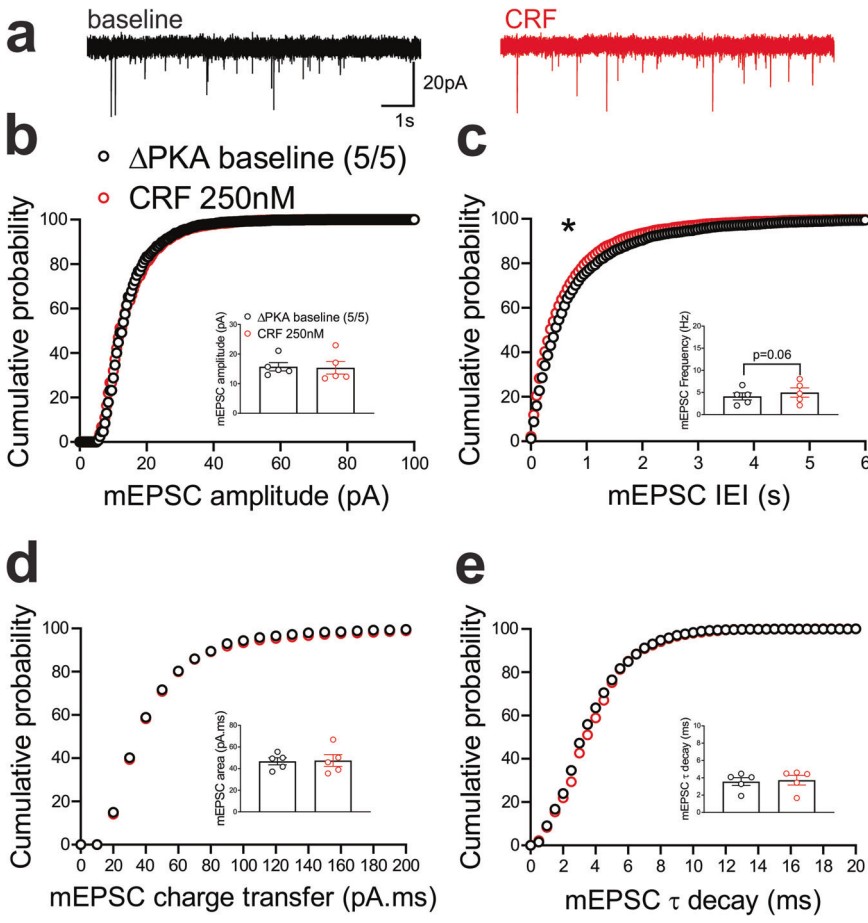

phosphorylation of GluA1 rectifying AMPARs by the genetic disruption of AKAP150 anchoring of PKA to GluA1. Note that the decrease in frequency of mEPSC in ΔPKA mice is likely postsynaptic, related to an increase in the number of silent synapses following the loss of AMPARs at the synapse[71] rather than a change in presynaptic glutamate release. Whether, the upregulation of calcium–calmodulin-dependent protein kinase II-regulated AKAP79/150 depalmitoylation that is important in AKAP150 removal from dendritic spine and structural LTD[72] is favored in ΔPKA mice is an open question.

Interestingly, disruption of PKA anchoring in ΔPKA mice is shown to impair an NMDAR-dependent LTD induced by prolonged LFS (1 Hz, 15 min similar to the LTD protocol in LHb) in CA1 hippocampal neurons of 2-week-old mice due to decreased S845 phosphorylation of CP-AMPARs in ΔPKA mice that prevents the AKAP150-PKA-dependent transient recruitment of CP-AMPARs to the synapse that is required for hippocampal LTD induction[9]. Interestingly, LFS can also induce eCB-mediated LTD in the LHb through increased activity of CP-AMPAR (as a major source of calcium) that further engages NMDARs to trigger retrograde eCB-LTD[59,60]. Given that the majority of AMPARs in the LHb are CP-AMPARs, a reduced level of CP-AMPARs in LHb neurons of ΔPKA mice could result in lower levels of depolarization and postsynaptic calcium needed for eCB production, thereby deficits in induction and expression of eCB-LTD. Moreover, since basal PKA phosphorylation of L-type calcium channels (LTCC) is necessary for depolarization-induced activation of LTCCs, the ΔPKA mutation could further diminish $Ca^{2+}$ influx through LTCC as an unopposed CaN activity can dephosphorylate LTCCs[14]. This in addition to the presence of fewer CP-AMPARs in LHb neurons of ΔPKA mice might result in further reduction in $Ca^{2+}$ influx, defective eCB production, and hence impaired eCB-LTD in LHb neurons.

GABA$_A$Rs at GABAergic synapses onto LHb neurons are mainly composed of a combination of the α1-3, β1 and γ1-2 subunits[73]. There is less

known about PKA-dependent regulation of GABA$_A$Rs in the LHb. Our previous study in VTA dopamine neurons suggests that activation of dopamine D2 receptors results in PKA inhibition that promotes AKAP150-CaN-mediated internalization of GABA$_A$R receptors and the expression of LTD at GABAergic synapses onto VTA dopamine neurons[29]. The expression of an inhibitory metabotropic glutamate receptor-dependent postsynaptic LTD at GABAergic synapses onto LHb neurons requires a PKC-dependent phosphorylation of the β2 receptor subunits of GABA$_A$Rs, reducing GABA$_A$R single-channel conductance[58]. This also excludes the possibility that a biased AKAP150-PKC signaling in the absence of AKAP-PKA association in ΔPKA mice could promote the basal increase in the conductance of GABA$_A$Rs in LHb neurons. Therefore, it is still an open question which AKAP150 associations with other binding partners could promote forward trafficking of GABA$_A$Rs in LHb neurons.

In addition to alterations of synaptic transmission and LTD by ΔPKA mutation, we observed a significant increase in LHb intrinsic excitability associated with higher input resistance and lower amplitude of mAHPs, mimicking the effects of exogenous CRF (as we observed in both mouse and rat LHb) and after a severe early like stress (i.e., maternal deprivation)[57]. However, the diminishing effects of exogenous CRF and maternal deprivation on mAHPs and the resultant hyperexcitability were due to the PKA-dependent decrease in the function and/or abundance of SK channels[57], a mechanism that is less likely to underlie ΔPKA mutation-induced LHb intrinsic plasticity. Afterhyperopolarizations including fAHPs and mAHPs are mediated by diverse types of potassium channels that repolarize the membrane to regulate and limit excessive neuronal excitability. In addition to SK channels, voltage gated K$^+$ channel 7 (Kv7, also known as M currents) contribute to mAHP in neurons[74]. Therefore, it is possible that genetic disruption of AKAP150 anchoring of PKA in ΔPKA favors AKAP150-anchored PKC and the resultant inhibition of M-type mAHPs[12] to increase LHb intrinsic excitability in ΔPKA mice, which could saturate and occlude

**Fig. 8 | CRF significantly suppressed presynaptic GABA release in the LHb of WT mice. a** Sample GABA$_A$R-mediated mIPSC traces from WT mouse before (black) and after CRF application (red, 250 nM) (calibration bars: 20 pA/1 s). Average bar graphs of mIPSC and cumulative probability plots of (**b**) amplitude, (**c**) frequency (inter-event interval), (**d**) charge transfer and (**e**) τ decay for all mIPSCs in WT mice before (black filled symbols) and after CRF (red filled symbols) (*n* = 3 cells from 3 mice). Two-tailed paired Student's *t*-tests and Kolmogorov-Smirnov tests, *\*p* < 0.05, *\*\*\*\*p* < 0.0001.

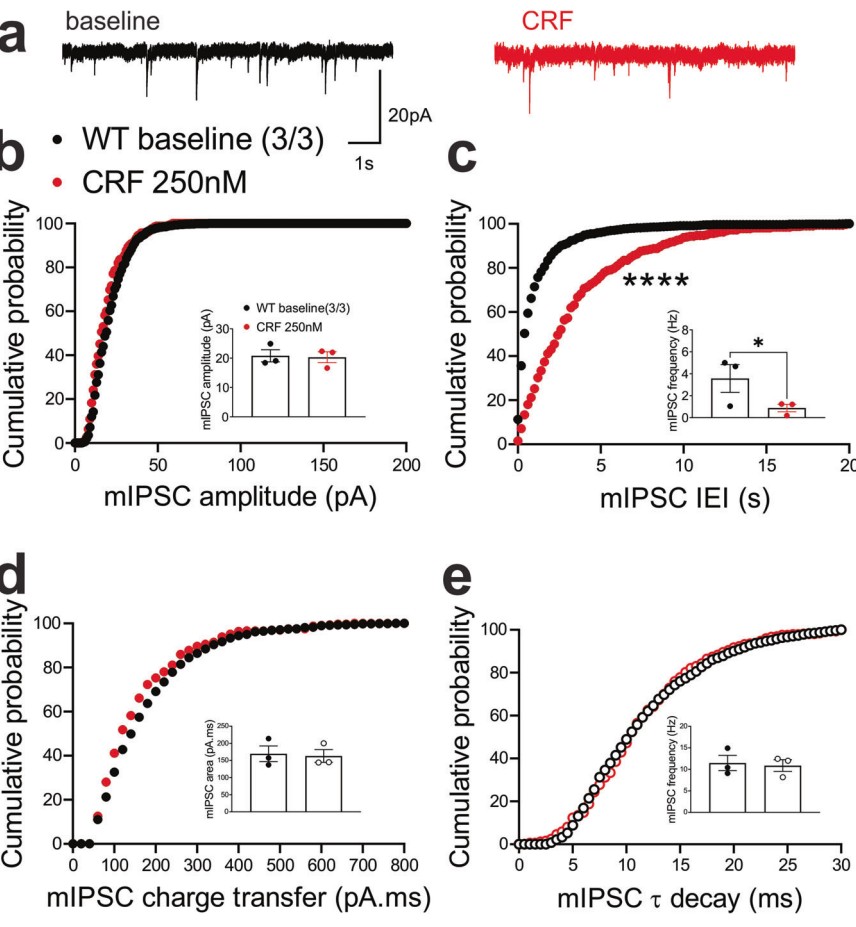

the excitatory actions of CRF on LHb intrinsic excitability. Consistent with this interpretation, it has been shown that activation of LHb M channels reduces LHb neuronal activity and blocks the anxiety-like phenotype in alcohol-withdrawn rats[75]. Given that that the majority of synaptic inputs to the LHb co-release glutamate and GABA[40], our observation of CRF-induced suppression of both presynaptic glutamate and GABA release in mouse LHb is not surprising as CB1 receptors are expressed on presynaptic terminals in the LHb where CB1 receptor activation by eCBs can reduce the probability of presynaptic glutamate and GABA release onto LHb neurons at distinct synaptic inputs to the LHb (e.g., lateral preoptic area) although the effect on presynaptic GABA release is assumed to be predominantly larger[76]. The ΔPKA mutation to some extent reduced the suppressing effects of CRF on presynaptic GABA release but also unmasked a small potentiating effect of CRF on presynaptic glutamate release. This could be due to decreased depolarization and/ or Ca$^{2+}$ influx from the fewer CP-AMPARs available at the synapse as well as the less effective influx of Ca$^{2+}$ from hypofunctional LTCCs in ΔPKA mice, which could in turn lead to dysregulation of eCB production that not only prevented the expression of eCB-LTD but also blunted the inhibitory effects of CRF on synaptic transmission by shifting excitation/inhibition balance to more excitation. Therefore, we assume that disruption of AKAP150 anchoring of PKA seems to promote LHb hyper-excitability through synaptic and intrinsic mechanisms that may relate to the lack of AKAP150-dependent PKA-mediated signaling as well as favoring unopposed non-PKA-mediated AKAP150 interactions. These concepts are briefly summarized in Supplementary Fig. 1, which depicts in schematic representing GABAergic and glutamatergic terminals innervating a spine and dendritic shaft of an LHb neuron. Sites in this synaptic complex where AKAP-mediated signaling plays a potential role in synaptic function and plasticity are indicated.

Overall, our study highlights the important and multifaceted impacts of AKAP150 anchoring of PKA in the regulation of glutamatergic

transmission and plasticity, and neuronal excitability of LHb neurons. Moreover, defects in AKAP150-mediated PKA anchoring under pathological processes could favor other, yet to-be-discovered, AKAP150 interactions in LHb neurons that promote LHb hyperactivity and dysregulate CRF neuromodulation within the LHb, reminiscent of the effects of a severe early life stress[57] and alcohol withdrawal[77,78]. Considering that human studies of polymorphisms of in the *AKAP5* gene indicate an essential role for this AKAP in emotional regulation and cognitive control of anger, aggression and impulsivity[20–23], it will be important to establish a direct link between the physiological and behavioral effects of such genetic variants or mutations of *AKAP5* with LHb circuit activity and LHb-regulation of emotionally motivated behaviors, behavioral impulsivity and aggression[35,38,79,80].

It is worth mentioning that there is a limitation to our genetic approach in which ΔPKA mutation is not confined to the LHb of ΔPKA mice. This raises the possibility that our current synaptic observations may not be solely limited to direct disruption of AKAP150-PKA complex signaling within the LHb of ΔPKA mice; but also involve dysregulation of AKAP150-PKA-mediated synaptic or neuromodulatory functions within brain areas that directly project to the LHb such as medial prefrontal cortex, VTA, amygdala and periaqueductal gray where AKAP150/PKA signaling is known to regulate synaptic mechanisms underlying impulsive, depressive-, aversive- and drug-related behaviors[8,15,27–29,33,34,81].

Future studies are further needed to increase our understanding of cell-type and input-specific neuroplasticity and neuromodulation through alterations in LHb AKAP150 complex interactions within LHb circuits in neurological and neuropsychiatric illnesses.

## Methods
### Animals
All experiments were carried out using 5-7wk old male WT (C57/Bl6) and ΔPKA mice in accordance with the National Institutes of Health (NIH)

**Fig. 9 | CRF suppressed presynaptic GABA release (to a lesser extent than that of WT) but also postsynaptically depressed GABAergic transmission in the LHb of ΔPKA mice. a** Sample GABA_AR-mediated mIPSC traces from ΔPKA mouse before (black) and after CRF application (red, 250 nM) (calibration bars: 20 pA/1 s). Average bar graphs of mIPSC and cumulative probability plots of (**b**) amplitude, (**c**) frequency (inter-event interval), (**d**) charge transfer and (**e**) τ decay for all mIPSCs in ΔPKA mice before (black open symbols) and after CRF (red open symbols) (n = 5 cells from 5 mice). Two-tailed paired Student's t-tests and Kolmogorov-Smirnov tests, *p < 0.05, ****p < 0.0001.

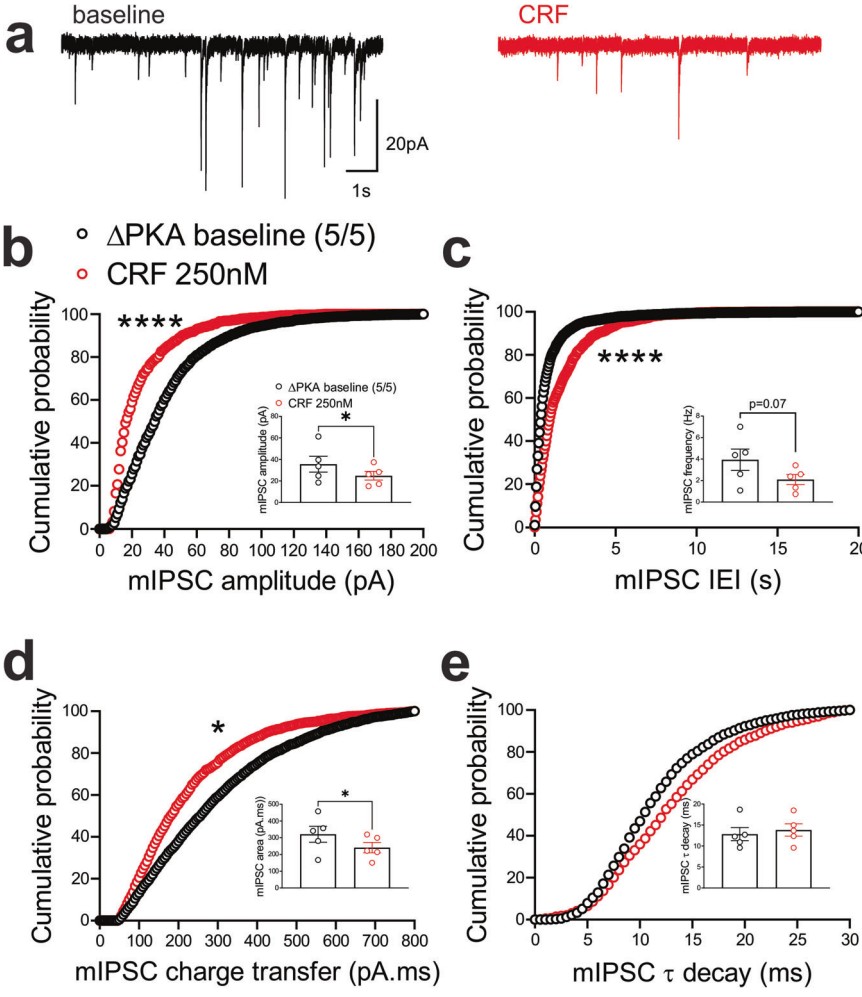

*Guide for the Care and Use of Laboratory Animals* and were approved by the Uniformed Services University and the University of Colorado (Denver) Institutional Animal Care and Use Committees. AKAP150ΔPKA knockin mice were generated as previously described[14]. We have complied with all relevant ethical regulations for animal use. Briefly, we constructed a targeting vector for the deletion of 30 bp encoding 709-LLIETASSLV-718 that was introduced into the single coding exon of an *Akap5* genomic DNA fragment subcloned from a C57BL/6 bacterial artificial clone. In this targeting vector, the ΔPKA mutation and a C-terminal myc-epitope tag were introduced with a loxP-flanked-neomycin resistance cassette inserted into the 3" flanking genomic DNA. Following electroporation of the targeting vector into a hybrid C57BL/6-129 embryonic stem cell. Targeted G418-resistant clones were screened for homologous recombinants by PCR. The positive clones were expanded and injected into blastocysts and transplanted into surrogate mothers. Chimeric F0 founders were born and subsequently bred to C57BL/6 to establish germ-line transmission. Intercrossing of F1 mice that were heterozygous for the ΔPKA mutation yielded F2 ΔPKA homozygotes. Mice were bred by Dell'Acqua laboratory at the University of Colorado Anschutz Medical Campus in Aurora, CO and shipped to Nugent laboratory at the Uniformed Services University in Bethesda, MD at ages between ~28–35 days old. After a week in quarantine at animal facility, mice were used for electrophysiology experiments. Given the concern regarding the possible higher stress levels of juvenile female mice in estrus phase during the shipment and the possibility of stress-induced hormonal dysregulation of LHb activity (a subpopulation of GABAergic interneurons are found to express estrogen receptor-alpha (ERα)(termed the GABAergic estrogen-receptive neuron or GERN)[39,41]), we decided to only use male mice for this study and plan for having a colony at

both sites for our future collaborative studies. Mice were grouped and housed in standard cages under a 12 hr/12 hr light-dark cycle with standard laboratory lighting conditions (lights on, 0600-1800), with ad libitum access to food and water. All procedures were conducted beginning 2–4 hr after the start of the light-cycle. All efforts were made to minimize animal suffering and reduce the number of animals used throughout this study.

### Slice preparation
For all electrophysiology experiments, several separate cohorts of WT/ ΔPKA mice were used. All mice were anesthetized with isoflurane, decapitated and brains were quickly dissected and placed into the ice-cold artificial cerebrospinal fluid (ACSF) containing (in mM): 126 NaCl, 21.4 NaHCO_3, 2.5 KCl, 1.2 NaH_2PO_4, 2.4 CaCl_2, 1.00 MgSO_4, 11.1 glucose, 0.4 ascorbic acid; saturated with 95% O_2−5% CO_2 as previously described[82]. Sagittal slices containing LHb were cut at 220 μm using a vibratome (Leica; Wetzler, Germany) and incubated in above prepared ACSF at 34 °C for at least 1 h prior to electrophysiological experiments. Slices were then transferred to a recording chamber and perfused with ascorbic-acid free ACSF at 28 °C.

### Electrophysiology
All whole-cell recordings were performed on LHb-containing slices using patch pipettes (3–6 MOhms) and a patch amplifier (MultiClamp 700B) under infrared-differential interference contrast microscopy. Data acquisition and analysis were carried out using DigiData 1440 A, pCLAMP 10 (Molecular Devices), Clampfit, Origin 2016 (OriginLab), Mini Analysis 6.0.3 (Synaptosoft Inc.) and GraphPad Prism 10. Signals were filtered at 3 kHz and digitized at 10 kHz. In all of our recordings, the cell input

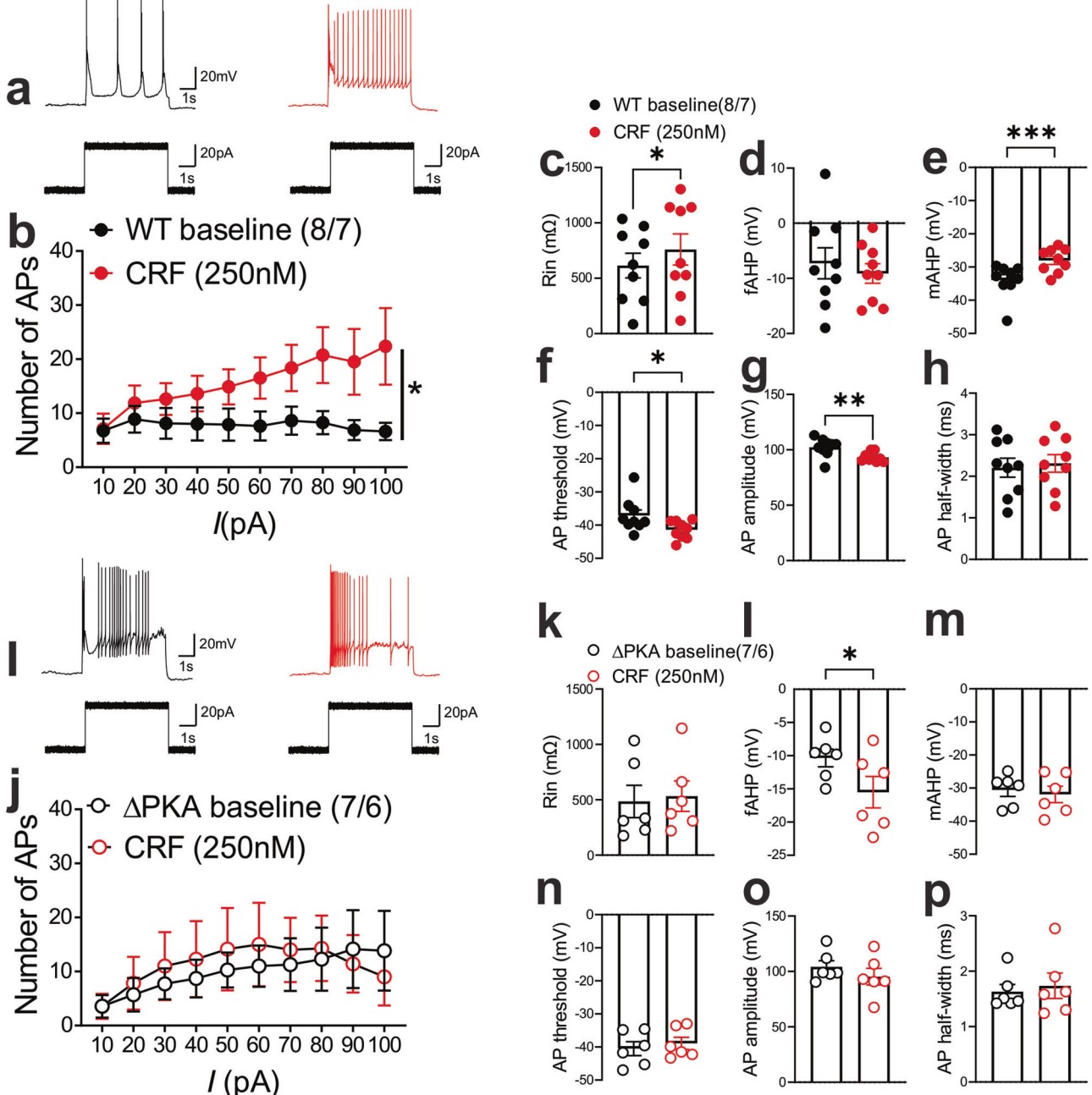

**Fig. 10 | Genetic disruption of AKAP150-anchored PKA occluded the effects of CRF on LHb excitability.** All recordings in this graph were performed with fast synaptic transmission blocked. **a–h** AP recordings in response to depolarizing current steps with representative AP traces (in response to a 40 pA current step) and corresponding measurements of Rin, fAHP, mAHP, AP threshold, AP amplitude and AP half-width before (baseline, black filled symbols), after CRF (250 nM, red filled symbols) bath application in LHb neurons from WT mice

($n = 8$ cells/7 mice). **i–p** AP recordings in response to depolarizing current steps with representative AP traces (in response to a 40 pA current step) and corresponding measurements of Rin, fAHP, mAHP, AP threshold, AP amplitude and AP half-width before (baseline, black open symbols), after CRF (250 nM, red open symbols) bath application in LHb neurons from ΔPKA mice ($n = 7$ cells from 6 mice). 2-way repeated-measures ANOVA and two-tailed paired Student's $t$-tests, $*p < 0.05$, $**p < 0.01$.

resistance and series resistance were monitored through the experiment and if these values changed by more than 10%, data were not included.

Whole-cell recordings of AMPAR-mediated miniature excitatory postsynaptic currents (mEPSCs) were isolated in ACSF perfused with GABA$_A$R antagonist picrotoxin (100 μM, Tocris-1128), NMDAR antagonist D-(-)-2-Amino-5- phosphonopentanoic acid (APV 50 μM, Tocris-0106) and tetrodotoxin (1 μM Tocris-1078) and internal solution containing 117 mM Cesium-gluconate, 2.8 mM NaCl, 5 mM MgCl$_2$, 2 mM ATP-Na$^+$, 0.3 mM GTP-Na$^+$, 0.6 mM EGTA, and 20 mM HEPES (pH 7.28, 275–280 mOsm). Whole-cell recordings of GABA$_A$R-mediated miniature

inhibitory postsynaptic currents (mIPSCs) were isolated in ACSF perfused with the AMPAR antagonist 6,7 dinitroquinoxaline-2,3-dione di-sodium salt (10 μM Tocris- 2312/10), strychnine (1 μM Tocris-2785) and tetrodotoxin (1 μM). Patch pipettes were filled with 125 mM KCl, 2.8 mM NaCl, 2 mM MgCl$_2$, 2 mM ATP-Na$^+$, 0.3 mM GTP-Na$^+$, 0.6 mM EGTA, and 10 mM HEPES (pH 7.28, 275–280 mOsm). For both mIPSCs and mEPSCs, LHb neurons were voltage-clamped at −70 mV and recorded over 10 sweeps, each lasting 50 seconds.

In some experiments, electrically-evoked AMPAR-mediated EPSCs were isolated and recorded using ACSF containing picrotoxin (100 μM).

The patch pipettes were filled with cesium-gluconate based solution as described above for mEPSC recordings. Cells were voltage-clamped at −70 mV, except during LTD protocol. Paired AMPAR-mediated EPSCs were stimulated at 0.1 Hz (100 ms) using a bipolar stainless steel stimulating electrode placed ~200–400 μm from the recording site in stria medularis in LHb slices. The stimulation intensity was adjusted so that the amplitude of synaptic responses ranged about ~50% of the maximum response. LTD was induced using low-frequency stimulation, LFS, 1 Hz for 15 min while LHb neurons were voltage-clamped at −40 mV.

To assess LHb intrinsic excitability and membrane properties, LHb slices were perfused with ascorbic-free ACSF and patched with potassium gluconate-based internal solution (130 mM K-gluconate, 15 mM KCl, 4 mM ATP-Na$^+$, 0.3 mM GTP-Na$^+$, 1 mM EGTA, and 5 mM HEPES, pH adjusted to 7.28 with KOH, osmolarity adjusted to 275 to 280 mOsm). LHb intrinsic excitability experiments were performed with fast-synaptic transmission blockade by adding the AMPAR antagonist 6,7 dinitroquinoxaline-2,3-dione di-sodium salt (10 μM), GABA$_A$R blocker picrotoxin (100 μM), and NMDAR antagonist D-(-)−2-Amino-5- phosphonopentanoic acid (50 μM) to the ACSF. LHb neurons were given increasingly depolarizing current steps at +10pA intervals ranging from +10pA to +100pA, allowing us to measure action potential (AP) generation in response to membrane depolarization (5 s duration). Current injections were separated by a 20 s interstimulus interval and neurons were kept at ~−65 to −70 mV with manual direct current injection between pulses. Resting membrane potential (RMP) was assessed immediately after achieving whole-cell patch configuration in the current clamp mode. Input resistance (Rin) was measured during a −50 pA step (5 s duration) and calculated by dividing the steady-state voltage response by the current-pulse amplitude (−50 pA) and presented as MOhms (MΩ). The number of APs induced by depolarization at each intensity was counted and averaged for each experimental group. As previously described[57] AP number, AP threshold, fast and medium after-hyperpolarization amplitudes (fAHP and mAHP), AP halfwidth, AP amplitude were assessed using Clampfit and measured at the current step that was sufficient to generate the first AP/s.

## Drugs

For all drug experiments, a within-subjects experimental design was employed. Stock solutions for CRF were prepared in distilled water and diluted (1:1000) to a final concentration in ACSF of 250 nM. Baseline recordings were first performed (depolarization-induced AP/mIPSC/mEPSC) for each neuron and then CRF (250 nM Tocris-1151) was added to the slice by the perfusate and response tested following 30–45 min of CRF application.

## Immunohistochemistry

Mice were anesthetized with an intraperitoneal injection containing ketamine (85 mg/kg) and xylazine (10 mg/kg) and perfused through the aorta with 200 ml of 1x phosphate buffered saline (PBS), followed by 250 ml of 4% paraformaldehyde (Santa Cruz). The brains were dissected and placed in 4% paraformaldehyde for 24 h and then cryoprotected by submersion in 20% sucrose for 3 d, frozen on dry ice, and stored at 70 °C until sectioned. Sections of the LHb were cut using a cryostat (Leica CM1900) and mounted on slides. Serial coronal sections (20 μm) of the midbrain containing the LHb (from −2.64 to −4.36 mm caudal to bregma (Paxinos and Watson, 2007) were fixed in 4% PFA for 5 min, washed in 1x PBS, and then blocked in 10% normal goat serum containing 0.3% Triton X-100 in 1x PBS for 1 h. Sections were incubated in goat anti- AKAP150 antibody (1:500, Santa Cruz Sc-6445) in carrier solution (5% normal goat serum in 0.1% Triton X-100 in 1x PBS) overnight at room temperature. After rinsing in 1x PBS, sections were incubated for 2 h in Alexa Fluor® 488 labeled chicken anti-goat IgG (diluted 1:200). Finally, sections were rinsed in 1x PBS, dried, and cover-slipped with prolonged mounting medium containing DAPI to permit visualization of nuclei. Background staining was assessed by omission of primary antibody in the immunolabeling procedure (negative control). Brain tissue sections of mice with a previously established presence of AKAP150 immunoreactive neurons (hippocampus, VTA) were also processed as positive control tissues. Images were captured using a Leica DMRXA Fluorescence microscope.

## Statistics and reproducibility

Values are presented as mean ± SEM. Statistical significance was determined using unpaired or paired two-tailed Student's t-test, two-way ANOVA, or repeated-measures ANOVA/mixed-effects ANOVA with Bonferroni post hoc analysis. The threshold for significance was set at *p < 0.05 for all analyses. The peak values of the evoked paired EPSCs were measured relative to the same baseline. A stable baseline value was considered in each sweep of paired pulses starting at 20-50 ms right before the emergence of the EPSC current using p-Clamp 10 software. The paired-pulse ratio (PPR) was calculated as the amplitude of the second EPSP divided by the amplitude of the first EPSC. The inverse square of the coefficient of variation (CV = SD/mean) was also used as the second measure for identifying the presynaptic expression of plasticity. For calculating the significance of EPSC amplitude changes after LTD induction protocol, amplitudes of EPSCs to the first pulse were used. Mini Analysis software was used to detect and measure mIPSCs and mEPSCs using preset detection parameters of mIPSCs and mEPSCs with an amplitude cutoff of 5 pA. The Kolmogorov–Smirnov test was performed for the statistical analyses of cumulative probability plots of mEPSCs and mIPSCs. All statistical analyses were performed using GraphPad Prism 10.

## Reporting summary

Further information on research design is available in the Nature Portfolio Reporting Summary linked to this article.

## Data availability

The raw data generated during this study that support the findings of this study are available on request from the corresponding authors. The raw data are not publicly available due to privacy or ethical restrictions; however, the analyzed dataset and supporting Information reported in the article's figures are included in Supplementary Data 1.

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

## Acknowledgements

The opinions and assertions contained herein are the private opinions of the authors and are not to be construed as official or reflecting the views of the Uniformed Services University of the Health Sciences or the Department of Defense or the Government of the United States. This work was supported by the National Institute of Mental Health (NIH/NIMH) and the National Institute of Neurological Disorders and Stroke (NIH/NINDS): Grants# R21 MH132136 to FSN and R01 MH123700 and R01 NS040701 to MLD. The funding agency did not contribute to writing this article or deciding to submit it.

## Author contributions

F.S.N. and M.L.D. designed the research; S.C.S., W.J.F., L.D.L., R.D.S. and C.B. performed electrophysiology; J.L.S. was responsible for breeding ΔPKA mice; S.G. K.M.G. and E.H.T. performed immunohistochemistry. S.C.S., W.J.F. and F.S.N. analyzed the data and prepared the figures; S.C.S., W.J.F., B.M.C., M.L.D. and F.S.N. wrote the initial draft of the manuscript. All authors critically reviewed the content and approved the final version of the manuscript for submission.

## Competing interests

The authors declare no conflict of interest. Fereshteh Nugent is an Editorial Board Member for *Communications Biology*, but was not involved in the editorial review of, nor the decision to publish this article.
