## [Peer Review File · Communications Biology]

Reviewers' comments:

Reviewer #1 (Remarks to the Author):

AKAP150 links PKA to various postsynaptic proteins including AMPARs. The authors used KI mice in which a deletion of about 18 residues near the C-terminus of AKAP150 abrogates binding of PKA to AKAP150. The lateral habenula (LHb) is a brain structure that functions to suppress motivation and reward learning. It mediates feedforward inhibition of different monoaminergic signaling in the brain including dopaminergic and norepinephrinergic nuclei that provide innervation throughout the brain. Accordingly, LHb has been implicated in depression and other psychiatric disorders, making the LHb clinically interesting.

Here the authors show that 5-7 week old AKAP150deltaPKA mice have under basal conditions a reduction in postsynaptic AMPAR responses (mEPSC amplitude) and an increase in GABA A receptor responses (mIPSC amplitude). They report that intrinsic excitability of LHb neurons is increased in these mice. Furthermore, CRF slightly decreased presynaptic glutamate release (mEPSC frequency) in WT mice but slightly increased it in AKAP150deltaPKA. CRF strongly decreased presynaptic GABA release (mIPSC frequency) in WT mice but only slightly in AKAP150deltaPKA.

The experiments are carefully executed and data are convincing and interesting. Thus, I only have a couple of modest concerns

Concerns

1. There seems to be a discrepancy between results reported in Fig. 5 versus Fig 10. In Fig 5A the excitability as measured by AP frequency in response to current injection is clearly increased whereas in Fig. 10A the increase under basal conditions (no CRF), which should be comparable to Fig 5A is much smaller – there seems to be a quantitative difference, which might affect the actual difference to WT mice.
2. What the abbreviation CRF means is not explained

Reviewer #2 (Remarks to the Author):

In this study the authors use an experimental mouse model that contains a mutation known as (Δ PKA) in which there is mutation in the A-kinase anchoring protein (AKAP150) that impairs its membrane anchoring of protein kinase A. As the authors indicate, this mutation selectively alters PKA anchoring without having significant effect on the anchoring of other enzymes/proteins by the AKAP150 complex, which is preferable to complete AKAP150 knockout models. These Δ PKA mutant mice are used to investigate the participation of AKAP150-anchored PKA on neuronal function and on the effects of CRF in the lateral habenula (LHb). To do this, the authors used whole cell electrophysiology in LHb brain slices and found significant changes in both glutamatergic and GABAergic synaptic transmission, as well as a deficit in endocannabinoid-mediated LTD in LHb neurons. Additionally, they report that the mutation was associated with an increase in LHb neuron excitability, and that this decreased the previously reported excitatory effects of CRF on LHb neurons. The experiments are well-designed and executed and the authors' interpretation of the data is generally accurate and straightforward. Therefore, I have only a few comments for the authors to consider.

1. Given that the LHb is involved in reward processing, drug addiction, impulsivity, and other psychiatric disorders, and that AKAP polymorphisms in humans are associated altered emotional processing and behavioral responses that are generally aligned with LHb function, the rationale for the present study is strong. However, the present work does not address potential changes in LHb

involvement in these behaviors that might result from altered AKAP150 function. Moreover, as the Δ PKA mutation is not restricted to the LHB in these mice, it is possible that any changes seen in an experimental behavioral context could not easily be ascribed to changes in LHB function. Although such detailed behavioral studies are clearly beyond the scope of the present work, it would be useful for the authors to discuss these issues as potential caveats of this work, or to more directly address them in the discussion.

2. As a continuation of the first point, the lack of selective knockin of the mutation in the LHB suggests that some of electrophysiological changes seen in LHB neurons might be referred or contributed by physiological changes occurring upstream from this structure. The altered synaptic properties, for example, could reflect changes resulting from the Δ PKA mutation that are "upstream" from the LHB in areas such as the hippocampus, where AKAP is known to modulate neuronal function. I think that this should at least be considered as a possibility.

3. Given the emphasis by NIH in promoting the use of sex as a biological variable, and the greater vulnerability of females to psychiatric disorders that may involve the LHB, it would be useful to justify in methods why only male mice were used in these studies.

4. The Δ PKA mutation is constitutively expressed, leaving open the possibility that some of the observations made in the present physiological study may result from compensatory changes in downstream signaling. Again, this possibility should be discussed or dismissed.

5. More information regarding the Δ PKA mice used in this study is necessary. For instance, there is no reference to their development in the animals section of the methods. Also, where were the mice obtained for this study?

6. Line 117: "In general, LHB dysfunction can mediate negative affective states, social deficits, risky decision-making and impulsivity (as shown in patients with depression, schizophrenia, Parkinson's disease and attention-deficit hyperactivity disorder, ADHD) 36, 49-57." However, the reference cited here do not seem to include studies of impulsive behavior.

Reviewers' comments:

We would like to thank both reviewers for their positive feedback and enthusiasm for this work and very much appreciate the constructive and valuable comments and suggestions. Based on some of the suggestions, we have revised the manuscript. Below we address all the concerns point-by-point for each reviewer.

Reviewer #1 (Remarks to the Author):

AKAP150 links PKA to various postsynaptic proteins including AMPARs. The authors used KI mice in which a deletion of about 18 residues near the C-terminus of AKAP150 abrogates binding of PKA to AKAP150. The lateral habenula (LHb) is a brain structure that functions to suppress motivation and reward learning. It mediates feedforward inhibition of different monoaminergic signaling in the brain including dopaminergic and norepinephrergic nuclei that provide innervation throughout the brain. Accordingly, LHb has been implicated in depression and other psychiatric disorders, making the LHb clinically interesting.

Here the authors show that 5-7 week old AKAP150deltaPKA mice have under basal conditions a reduction in postsynaptic AMPAR responses (mEPSC amplitude) and an increase in GABA A receptor responses (mIPSC amplitude). They report that intrinsic excitability of LHb neurons is increased in these mice. Furthermore, CRF slightly decreased presynaptic glutamate release (mEPSC frequency) in WT mice but slightly increased it in AKAP150deltaPKA. CRF strongly decreased presynaptic GABA release (mIPSC frequency) in WT mice but only slightly in AKAP150deltaPKA.

The experiments are carefully executed and data are convincing and interesting. Thus, I only have a couple of modest concerns

We thank the reviewer for finding our study interesting and very much appreciate the constructive comments.

Concerns

1. There seems to be a discrepancy between results reported in Fig. 5 versus Fig 10. In Fig 5A the excitability as measured by AP frequency in response to current injection is clearly increased whereas in Fig. 10A the increase under basal conditions (no CRF), which should be comparable to Fig 5A is much smaller – there seems to be a quantitative difference, which might affect the actual difference to WT mice.

We completely agree with your careful observation, but the reason for this difference is that Fig. 10 only represents LHb neurons with matched baseline before and after CRF bath application. While in Fig. 5 we included both the baseline intrinsic excitability of the neurons from Fig. 10 and an additional number of neurons that we were only able to record their baseline activity without CRF. This gave us a better sample size to show the relative difference in baseline intrinsic excitability of LHb neurons from WT versus Δ PKA mice. We added this detail to the Results section which is also highlighted.

In a subset of the neurons represented from in Fig. 5A, we were also able to examine the effects of CRF bath application on LHb intrinsic excitability in slices from WT and Δ PKA mice (i.e., intrinsic excitability recordings before and after CRF bath application).

2. What the abbreviation CRF means is not explained

We thank you for catching this. We have now spelled this out.

Reviewer #2 (Remarks to the Author):

In this study the authors use an experimental mouse model that contains a mutation known as (Δ PKA) in

which there is mutation in the A-kinase anchoring protein (AKAP150) that impairs its membrane anchoring of protein kinase A. As the authors indicate, this mutation selectively alters PKA anchoring without having significant effect on the anchoring of other enzymes/proteins by the AKAP150 complex, which is preferable to complete AKAP150 knockout models. These Δ PKA mutant mice are used to investigate the participation of AKAP150-anchored PKA on neuronal function and on the effects of CRF in the lateral habenula (LHb). To do this, the authors used whole cell electrophysiology in LHb brain slices and found significant changes in both glutamatergic and GABAergic synaptic transmission, as well as a deficit in endocannabinoid-mediated LTD in LHb neurons. Additionally, they report that the mutation was associated with an increase in LHb neuron excitability, and that this decreased the previously reported excitatory effects of CRF on LHb neurons. The experiments are well-designed and executed and the authors' interpretation of the data is generally accurate and straightforward. Therefore, I have only a few comments for the authors to consider.

We appreciate this reviewer's thorough review of our manuscript and finding our study well-designed and of interest as well as the constructive comments.

1. Given that the LHb is involved in reward processing, drug addiction, impulsivity, and other psychiatric disorders, and that AKAP polymorphisms in humans are associated altered emotional processing and behavioral responses that are generally aligned with LHb function, the rationale for the present study is strong. However, the present work does not address potential changes in LHB involvement in these behaviors that might result from altered AKAP150 function. Moreover, as the Δ PKA mutation is not restricted to the LHb in these mice, it is possible that any changes seen in an experimental behavioral context could not easily be ascribed to changes in LHb function. Although such detailed behavioral studies are clearly beyond the scope of the present work, it would be useful for the authors to discuss these issues as potential caveats of this work, or to more directly address them in the discussion.

We completely agree with the reviewer's point of view and their excellent suggestion by now including some interpretation for human studies of AKAP polymorphism in the context of behavioral involvement of LHb in aggression and impulsivity seen in carriers of AKAP variants. In addition, we now discuss the limitations of our genetic approach in which the Δ PKA mutation is not limited to the LHb, hence the possibility that dysregulation of AKAP150-PKA signaling in brain regions that project to the LHb also may underlie the physiological changes in the LHb function. We have now included these in the discussion section.

2. As a continuation of the first point, the lack of selective knockin of the mutation in the LHb suggests that some of electrophysiological changes seen in LHb neurons might be referred or contributed by physiological changes occurring upstream from this structure. The altered synaptic properties, for example, could reflect changes resulting from the Δ PKA mutation that are "upstream" from the LHb in areas such as the hippocampus, where AKAP is known to modulate neuronal function. I think that this should at least be considered as a possibility.

We thank the reviewer for this excellent suggestion and we addressed this in the previous comment by revising our Discussion. However, since there is no evidence for direct connection between the hippocampus and the LHb¹, we decided to only highlight the known brain regions that project to the LHb including the mPFC, VTA, PAG and amygdala with the known function of AKAP150/PKA signaling in these regions in the context of impulsive, depressive-, aversive- and drug-related behaviors.

3. Given the emphasis by NIH in promoting the use of sex as a biological variable, and the greater vulnerability of females to psychiatric disorders that may involve the LHb, it would be useful to justify in methods why only male mice were used in these studies.

We completely agree with you and include this limitation and justification in the Methods section. In fact, we have seen different levels of LHb neuronal excitability in male versus female LHb which is interesting given that a subpopulation of GABAergic interneurons in medial LHb are identified that express VGAT

and estrogen receptor-alpha (ER α) (termed the GABAergic estrogen-receptive neuron or GERN)^{2,3}. We added this explanation to the method section.

“Mice were bred by Dell’Acqua laboratory at the University of Colorado Anschutz Medical Campus in Aurora, CO and shipped to Nugent laboratory at the Uniformed Services University in Bethesda, MD at ages between ~28-35 days old. After a week in quarantine at animal facility, mice were used for electrophysiology experiments. Given the concern regarding the possible higher stress levels of juvenile female mice in estrus phase during the shipment and the possibility of stress-induced hormonal dysregulation of Lhb activity (a subpopulation of GABAergic interneurons are found to express estrogen receptor-alpha (ER α) (termed the GABAergic estrogen-receptive neuron or GERN)^{2,3}), we decided to only use male mice for this study and plan for having the colony at both sites for our future collaborative studies.”

4. The Δ PKA mutation is constitutively expressed, leaving open the possibility that some of the observations made in the present physiological study may result from compensatory changes in downstream signaling. Again, this possibility should be discussed or dismissed.

This is a valid point which we have discussed this possibility in multiple parts of the manuscript which may have been missed by the reviewer. We also think that disruption of AKAP150-PKA interaction in Δ PKA mutation may favor other AKAP150 partners and their downstream signaling as a compensatory mechanism. Specifically, this was discussed for the possible interaction of AKAP150 with PKC in the absence of AKAP-PKA interaction. We have highlighted those sections for the reviewer.

5. More information regarding the Δ PKA mice used in this study is necessary. For instance, there is no reference to their development in the animals section of the methods. Also, where were the mice obtained for this study?

We have included information on the generation of Δ PKA mice by the Dell’Acqua laboratory (previously published in Murphy et al., 2014; Sanderson et al., 2016^{4,5}) and the information regarding the transfer of the mice between institutes in Methods section.

6. Line 117: “In general, Lhb dysfunction can mediate negative affective states, social deficits, risky decision-making and impulsivity (as shown in patients with depression, schizophrenia, Parkinson’s disease and attention-deficit hyperactivity disorder, ADHD) 36, 49-57.” However, the reference cited here do not seem to include studies of impulsive behavior.

There are two references for Lhb studies in ADHD which implies a role for Lhb dysfunction in ADHD which is characterized by inattention, hyperactivity, and impulsivity^{6,7}.

References:

1. Baker, P.M., Rao, Y., Rivera, Z.M.G., Garcia, E.M. and Mizumori, S.J.Y. (2019). Selective Functional Interaction Between the Lateral Habenula and Hippocampus During Different Tests of Response Flexibility. *Frontiers in molecular neuroscience* 12, 245.
2. Zhang, L., Hernandez, V.S., Swinny, J.D., Verma, A.K., Giesecke, T., Emery, A.C., Mutig, K., Garcia-Segura, L.M. and Eiden, L.E. (2018). A GABAergic cell type in the lateral habenula links hypothalamic homeostatic and midbrain motivation circuits with sex steroid signaling. *Translational psychiatry* 8, 50.
3. Zhang, L., Hernandez, V.S., Vazquez-Juarez, E., Chay, F.K. and Barrio, R.A. (2016). Thirst Is Associated with Suppression of Habenula Output and Active Stress Coping: Is there a Role for a Non-canonical Vasopressin-Glutamate Pathway? *Frontiers in neural circuits* 10, 13.
4. Murphy, J.G., Sanderson, J.L., Gorski, J.A., Scott, J.D., Catterall, W.A., Sather, W.A. and Dell’Acqua, M.L. (2014). AKAP-Anchored PKA Maintains Neuronal L-type Calcium Channel Activity and NFAT Transcriptional Signaling. *Cell reports*.

5. Sanderson, J.L., Gorski, J.A. and Dell'Acqua, M.L. (2016). NMDA Receptor-Dependent LTD Requires Transient Synaptic Incorporation of Ca²⁺-Permeable AMPARs Mediated by AKAP150-Anchored PKA and Calcineurin. *Neuron* 89, 1000-1015.
6. Lee, Y.-A. and Goto, Y. (2021). The Habenula in the Link Between ADHD and Mood Disorder. *Frontiers in Behavioral Neuroscience* 15.
7. Arfuso, M., Salas, R., Castellanos, F.X. and Krain Roy, A. (2021). Evidence of Altered Habenular Intrinsic Functional Connectivity in Pediatric ADHD. *J Atten Disord* 25, 749-757.

REVIEWERS' COMMENTS:

Reviewer #1 (Remarks to the Author):

The authors fully addressed the comments.

Reviewer #2 (Remarks to the Author):

The authors have done a really nice job of responding to the comments made in the prior review. All of my comments have been addressed and I appreciate their attention to these detailed comments.